# Factors associated with psychological distress during the coronavirus disease 2019 (COVID-19) pandemic on the predominantly general population: A systematic review and meta-analysis

**Yeli Wang[1☯], Monica Palanichamy Kala[2☯], Tazeen H. Jafar[1,3,4]***

1 Program in Health Services and Systems Research, Duke-NUS Medical School, Singapore, Singapore,
2 Duke-NUS Medical School, Singapore, Singapore, 3 Department of Renal Medicine, Singapore General Hospital, Singapore, Singapore, 4 Duke Global Health Institute, Duke University, Durham, North Carolina, United States of America

☯ These authors contributed equally to this work.
* tazeen.jafar@duke-nus.edu.sg

**Data Availability Statement:** All relevant data are within the paper and its Supporting Information files.

## Abstract

### Background

The Coronavirus Disease 2019 (COVID-19) outbreak has escalated the burden of psychological distress. We aimed to evaluate factors associated with psychological distress among the predominantly general population during the COVID-19 pandemic.

### Methods

We searched PubMed, EMBASE, Scopus, Cochrane Library, PsycINFO, and World Health Organization COVID-19 databases (Dec 2019–15 July 2020). We included cross-sectional studies that reported factors associated with psychological distress during the COVID-19 pandemic. Primary outcomes were self-reported symptoms of anxiety and depression. Random-effects models were used to pool odds ratios (OR) and 95% confidence intervals (CI). The protocol was registered in PROSPERO (#CRD42020186735).

### Findings

We included 68 studies comprising 288,830 participants from 19 countries. The prevalence of anxiety and depression was 33% (95% CI: 28%-39%) and 30% (26%-36%). Women versus men (OR: 1.48 [95% CI: 1.29–1.71; $I^2$ = 90.8%]), younger versus older (< versus ≥35 years) adults (1.20 [1.13–1.26]; $I^2$ = 91.7%), living in rural versus urban areas (1.13 [1.00–1.29]; $I^2$ = 82.9%), lower versus higher socioeconomic status (e.g. lower versus higher income: 1.45 [1.24–1.69; $I^2$ = 82.3%]) were associated with higher anxiety odds. These factors (except for residential area) were also associated with higher depression odds. Furthermore, higher COVID-19 infection risk (suspected/confirmed cases, living in hard-hit areas,

**Funding:** THJ receives funding from the National Medical Research Council, Singapore. The funder played no role in the study design, data collection and analysis, decision to publish, or preparation of the manuscript.

**Competing interests:** The authors have declared that no competing interests exist.

having pre-existing physical or mental conditions) and longer media exposure were associated with higher odds of anxiety and depression.

## Interpretation

One in three adults in the predominantly general population have COVID-19 related psychological distress. Concerted efforts are urgently needed for interventions in high-risk populations to reduce urban-rural, socioeconomic and gender disparities in COVID-19 related psychological distress.

## Introduction

The Coronavirus Disease 2019 (COVID-19) outbreak has posed serious threats to public health across the globe. As of 23 August 2020, over 23 million confirmed cases and more than 800,000 deaths have been reported in 216 countries worldwide [1]. The unparalleled rate of transmission and the interruption of routine life by the institution of containment interventions (e.g. lockdown, quarantine, social distancing) has resulted in an adverse psychological impact on the mental well-being of populations across the globe [2–5]. A recent meta-analysis including studies from 17 countries conducted during the COVID-19 pandemic showed that 32% and 27% of the general population have symptoms of depression and anxiety, respectively [6], which sharply increased from the corresponding prevalence of 4.4% and 3.6% estimated in 2015 globally [7].

However, factors associated with the increased susceptibility to psychological distress during the COVID-19 pandemic are not well known. A few recent studies found that women [8–12], individuals with lower socioeconomic status (SES) (lower levels of education and income, and unemployment) [8, 13–19], residing in rural areas [13, 18, 19], and those with higher risk of COVID-19 infection [15, 20–22] have higher prevalence of depression and anxiety compared to their respective counterparts. However, results have not been entirely consistent, and some other studies did not observe the above-mentioned associations [8, 10, 13, 16, 20, 21]. Although a few meta-analyses have been conducted to investigate the prevalence of psychological distress related to the COVID-19 pandemic, studies on determinants of psychological distress have largely focused on healthcare workers [6, 23–26]. Systematic reviews on factors associated with psychological distress in the general population during the COVID-19 pandemic have not been reported. Understanding these factors is of significant clinical and public health importance worldwide for the risk stratification and designing psychosocial intervention programs. Studies have shown that psychosocial interventions are beneficial for the prevention and treatment of anxiety and depression and therefore could reduce the related morbidity [27–29]. Given the rapidly developing situation of the COVID-19, policy makers across the globe need the best evidence urgently to guide resource planning and targeted interventions for the public.

Therefore, we conducted a systematic review and meta-analysis to explore factors associated with psychological distress among the predominantly general population including high-risk or vulnerable patients with particular focus on gender, age, rural residence, and SES strata. We hypothesize that women, older adults, individuals residing in rural versus urban areas, and those of lower SES strata are associated with higher odds of psychological distress during the COVID-19 pandemic.

## Materials and methods

### Search strategy

To conduct the current systematic review and meta-analysis, we followed the Preferred Reporting Items for Systematic reviews and Meta-Analyses (PRISMA) [30] and Meta-analysis Of Observational Studies in Epidemiology (MOOSE) [31] guidelines. We conducted a systematic search on PubMed, EMBASE, Scopus, Cochrane Library, PsycINFO, and the World Health Organization (WHO) COVID-19 database (from Dec 2019 to 15 July 2020). We also manually searched the references of relevant reviews [6, 23–25, 32–37]. We did not assess grey literature sources. The full list of search terms can be found in the Supplemental Material. In brief, we used a combination of terms relating to psychological distress (e.g. anxiety, depression, stress, distress, post-traumatic stress, insomnia) and COVID-19 (e.g. COVID, 2019-ncov, sars-cov-2, novel coronavirus, severe acute respiratory syndrome coronavirus 2). MESH and Emtree terms with explosion of narrower terms were used to broaden search results. Prior to the literature search, we registered our study protocol with the National Institute for Health Research International prospective register of systematic reviews (PROSPERO, #CRD42020186735) [38].

### Selection criteria

Two investigators (M.P.K. and Y.W.) independently performed the search and assessed all articles for eligibility, and any discrepancy was resolved after discussing with a third investigator (T.H.J.). Articles were considered for inclusion if: 1) authors reported risk estimates (odds ratio [OR] and 95% confidence interval [CI]) of factors associated with higher odds of self-reported psychological distress (e.g. anxiety, depression, distress, stress, post-traumatic stress, and insomnia) using standardized and validated psychometric tools; 2) studies reported at least one of the pre-defined factors: gender, age, rural residence, and SES strata (education, income, and employment status); and 3) articles were original, peer-reviewed cross-sectional studies and published in English or Chinese languages. Articles were excluded if they: 1) were not relevant (not using pre-defined factors as the exposure or psychological distress of COVID-19 as the outcome); 2) did not report the OR of factors (e.g. studies using linear regression analyses) or associated 95% CI; 3) were animal or experimental studies, reviews, or meta-analyses; 4) were conducted exclusively among healthcare professionals. Eligibility was assessed by first screening titles and abstracts, followed by full-text reviews.

The following summary estimates of included articles were extracted in an excel sheet using pre-defined formats: study characteristics (study name, authors, journal, publication year, study design, study location, sample size), population characteristics (gender, mean/median age or age range), psychological distress assessment methods (psychometric tools and their thresholds), analytical strategies (statistical model, covariates) and results (risk estimates [ORs] and 95% CIs). We extracted risk estimates from the fully adjusted multivariable models when available. If the information was unclear or the full-text paper unavailable, we contacted authors for inquiry.

### Statistical analysis

For this meta-analysis, primary outcomes were anxiety and depression, and secondary outcomes were distress, stress, post-traumatic stress, and insomnia. ORs from logistic regression models were considered as risk estimates. To improve consistency between studies, data was transformed using the same reference group. When risk estimates were reported in subgroups instead of the total population, a within-study risk estimate combining multiple subgroups

was attained using a fixed-effect analysis [39]. When data were available for three or more studies, ORs and 95% CIs were pooled by the DerSimonian and Laird random effects model using a variation on the inverse-variance method to account for differences in the effect size (heterogeneity) among included studies [40]. We also pooled the prevalence of anxiety and depression from studies with available information. The between-study heterogeneity was evaluated with Cochrane Q statistic ($P<0.10$ indicates statistical significance) and $I^2$ statistic ($>50\%$ indicates possible heterogeneity) [41, 42]. We conducted meta-regression ($P<0.05$ indicates statistical significance) and stratified analyses by study locations and different instruments/cut-off points to evaluate the potential influence of geographic differences and variations in psychometric instruments and cut-off points on the results. In the sensitivity analyses, we further repeated the analyses excluding studies containing high-risk or vulnerable populations, or studies using random sampling techniques and thus containing a small subset of healthcare workers. When data were available for ten or more studies, the publication bias was assessed by Egger's regression ($P<0.05$ indicates statistical significance) and the funnel plot asymmetry [43]. If the potential publication bias exists, the trim-and-fill method was further used to assess the effect of publication bias [44]. Stata version 14.0 (Stata Corp, College Station, Texas) was used for all data analyses.

We used the Joanna Briggs Institute tool for cross-sectional studies to assess the quality of included studies on assessing psychological distress [45], which was also used for assessing the burden of psychological distress of previous infectious outbreaks (e.g. the severe acute respiratory syndrome [SARS], Ebola, H1N1) [23]. The scale included the following three domains: 1) appropriate selection of the population (representativeness of the sample, clear inclusion criteria), 2) comparability of the groups (identify and control for potential confounding factors, appropriate statistical analysis), and 3) valid and reliable measurement of exposures and outcomes. Overall, the studies were awarded a maximum of eight points, and a score of seven and above indicated high study quality [23].

If included studies reported the prevalence of psychological distress among patients with and without COVID-19, we further calculated the attributable risk of psychological distress due to COVID-19 by the formula $\frac{R1/R0-1}{R1/R0}$, where $R1/R0$ is the causal risk ratio that measures the risk under exposure (COVID-19) [46].

## Results

Our initial search identified 19,083 citations from six databases. After removing duplicates and screening for title, abstract and full text, we included 68 studies in the current meta-analysis (Fig 1). Four articles were published in Chinese [8, 20, 21, 47] and 64 were published in English [9, 11–19, 22, 26, 48–99]. The detailed characteristics of the included publications are shown in Table 1. Among the included studies, 41 were from the WHO Western Pacific Region (39 from mainland China [8, 9, 12–14, 18–22, 26, 47, 49–53, 57, 58, 60, 63, 65, 73, 74, 77–82, 86, 87, 89, 90, 92–94, 96, 97], one from Japan [66], and one from Vietnam [70]), 16 were from the European Region (six from Italy [11, 48, 55, 62, 68, 84], two from UK [16, 75], two from Spain [83, 91], two from Turkey [17, 88], one from Slovenia [85], one from Albania [61], one from France [59], and one from Ireland [72]), four were from the Region of the Americas (three from US [15, 56, 69], and one from Colombia [71]), four were from the Eastern Mediterranean Region (one from Iran [98], one from Israel [64], one from Saudi Arabia [99], and one from Jordan [67]), two were from the South-East Asia Region (India [54, 95]), and one was from the African Region (Tunisia [76]). Before May 2020, majority of the studies were from the Western Pacific Region (e.g. China and Vietnam); studies from other WHO regions started to emerge from May onwards (Fig 2). A total of 288,830 participants were included in the meta-analysis.

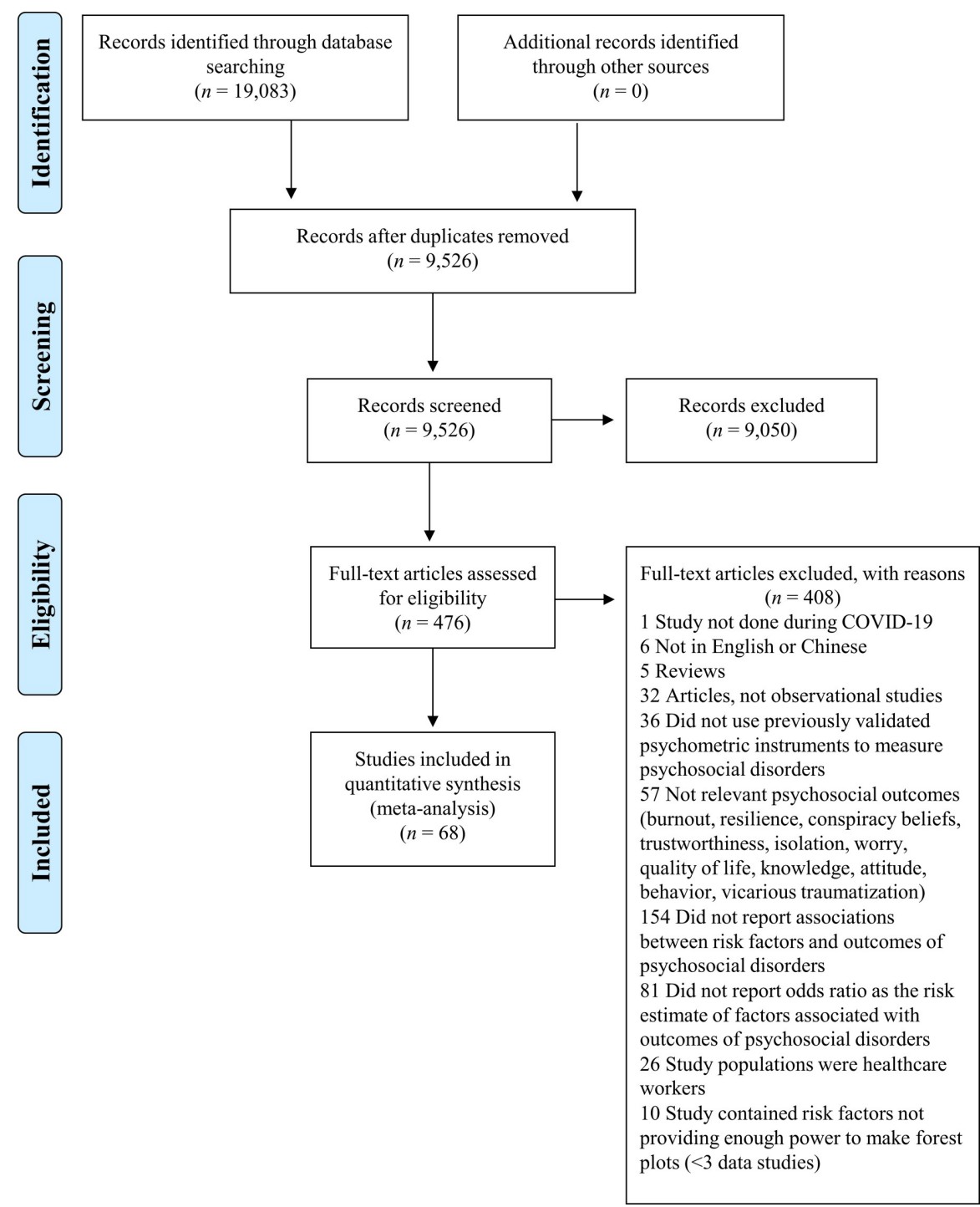

**Fig 1. Study selection of the meta-analysis.**

The majority of studies ($n$ = 62; 91.2%) were conducted among people aged 18 years or older and 59.3% of the participants were women. The study quality ranged between six and eight (fair to high), where most studies had high quality ($n$ = 58, 85.3%) indicated by a score of seven or higher. The most common problem affecting the study quality was accounting for confounding factors. The detailed study quality assessment is shown in S1 Table in S1 File.

**Table 1. Baseline characteristics of studies included in the meta-analysis.**

| Author, study location | Participant information | Age (mean, median or range) | Gender Female (%) | Sample size | Psychological distress | Cut-off points | Prevalence of psychological distress (%) | Psychometric instruments | Study quality |
|---|---|---|---|---|---|---|---|---|---|
| Mazza, Italy [68] | General population | 32.94 (13.2) | 71.7 | 2763 | Anxiety | >10.3 | 18.7 | DASS-21 | 8 |
| | | | | | Depression | >15.0 | 32.8 | | |
| | | | | | Stress | >18.3 | 27.2 | | |
| Moccia, Italy [48] | General population | Range: 18–75 | 59.6 | 500 | Distress | >19 | 38 | K10 | 8 |
| Li, China [49] | General population | Not reported | 66.7 | 5033 | Anxiety | >8 | 20.4 | GAD-7 | 6 |
| | | | | | Depression | >8 | | PHQ-9 | |
| Li, China [8] | Patients: COVID-19 | 36 (15) | 46 | 76 | Anxiety | ≥7 | 47.4 | HAM-A | 8 |
| | | | | | Depression | ≥4 | 30.3 | HAM-D | |
| Huang, China [57] | General population | 35.3 (5.6) | 54.6 | 7236 | Anxiety | ≥9 | 35.1 | GAD-7 | 8 |
| | | | | | Depression | >28 | 20.1 | CES-D | |
| | | | | | Insomnia | >7 | 18.2 | PSQI | |
| Li, China [58] | General population | 34.46 (9.62) | 63 | 3637 | Insomnia | >7 | 33.7 | ISI | 6 |
| Özdin, Turkey [17] | General population | 37.16 (10.31) | 49.3 | 343 | Anxiety | >7 | 45.1 | HADS | 6 |
| | | | | | Depression | >10 | 23.6 | | |
| Zhang, China [12] | General population: HCWs + NHCWs | Not reported | 64.2 | 2182 | Anxiety | ≥3 | 9.5 | GAD-2 | 8 |
| | | | | | Depression | ≥3 | 8.5 | PHQ-2 | |
| | | | | | Insomnia | >8 | 30.5 | ISI | |
| Gao, China [13] | General population | 32.3 (10) | 67.7 | 4827 | Anxiety | ≥10 | 22.6 | GAD-7 | 8 |
| | | | | | Depression | <13 | 48.3 | WHO-5 | |
| Xie, China [100] | General population: Children | Not reported | 43.3 | 1784 | Anxiety | NA | 18.9 | SCARED | 7 |
| | | | | | Depression | NA | 22.6 | CDI-S | |
| Chang, China [18] | General population: College students | 20 (19, 22) | 63 | 3881 | Anxiety | ≥6 | 26.6 | GAD-7 | 8 |
| | | | | | Depression | ≥5 | 21.2 | PHQ-9 | |
| Ni, China [20] | General population | Not reported | 60.8 | 1577 | Anxiety | ≥3 | 23.9 | GAD-2 | 7 |
| | | | | | Depression | ≥3 | 19.2 | PHQ-2 | |
| Nguyen, Vietnam [70] | Patients: COVID-19 and other diseases | 44.4 (17) | 55.7 | 3947 | Depression | ≥10 | 7.4 | PHQ-9 | 8 |
| Zhou, China [22] | General population: Adolescents | 16 (12, 18) | 53.5 | 8079 | Anxiety | ≥5 | 37.4 | GAD-7 | 8 |
| | | | | | Depression | ≥5 | 43.7 | PHQ-9 | |
| Iasevoli, Italy [101] | Patients: Mental illness | Range: 18–70 | Not reported | 461 | Anxiety | >10 | Not reported | GAD-7 | 7 |
| | | | | | Depression | >15 | | PHQ-9 | |
| | | | | | Stress | >26 | | PSS | |
| Hao, China [102] | Patients: Epilepsy | 29.3 (11.6) | 52.4 | 504 | Distress | >12 | 13.1 (severe) | K6 | 8 |
| Wang, China [96] | General population | 34 (12) | 55.5 | 600 | Anxiety | ≥50 | 6.3 | SAS | 7 |
| | | | | | Depression | ≥53 | 17.2 | SDS | |
| Cao, China [19] | General population: College students | Not reported | 69.7 | 7143 | Anxiety | ≥9 | 24.9 | GAD-7 | 8 |
| Chen, China [9] | General population: Children and Adolescents | Range: 6–15 | 48.7 | 1036 | Anxiety | ≥25 | 18.9 | SCARED | 7 |
| | | | | | Depression | ≥15 | 11.8 | DSRS-C | |
| Guo, China [52] | General population | Not reported | 52.4 | 2441 | Depression | ≥21 | 72.6 | CESD | 8 |
| | | | | | PTSS | | 79.6 | PTSD DSM-5 | |
| | | | | | Insomnia | ≥7 | 20.6 | PSQI | |
| Smith, UK [16] | General population | Not reported | 63.3 | 932 | Anxiety | ≥16 | Not reported | BAI | 8 |
| | | | | | Depression | ≥20 | | BDI | |

*(Continued)*

**Table 1.** (Continued)

| Author, study location | Participant information | Age (mean, median or range) | Gender Female (%) | Sample size | Psychological distress | Cut-off points | Prevalence of psychological distress (%) | Psychometric instruments | Study quality |
|---|---|---|---|---|---|---|---|---|---|
| Liu, US [56] | General population | 24.5 (18–30.9) | 81.3 | 898 | Anxiety | ≥10 | 45.4 | GAD-7 | 8 |
| | | | | | Depression | ≥10 | 43.3 | PHQ-8 | |
| | | | | | PTSD | ≥45 | 31.8 | PCL-C | |
| Costantini, Italy [62] | General population | 46.49 (13.58) | 58 | 329 | Distress | >49 | 25.2 | CPDI | 6 |
| Pedrozo-Pupo, Colombia [71] | General population | 43.9 (12.4) | 61.8 | 406 | Stress | ≥25 | 14.3 (high) | PSS-10-C | 7 |
| Chen, China [47] | General population | 32.3 (10) | 67.7 | 4827 | Anxiety | ≥10 | 55.3 | GAD-7 | 8 |
| Gómez-Salgado, Spain [91] | General population | 40.26 (13.18) | 74 | 4180 | Distress | ≥3 | 72 | GHQ-12 | 8 |
| Forte, Italy [11] | General population | 30 (11.5) | 74.6 | 2291 | Anxiety | ≥55 | 37.2 | STAI | 7 |
| | | | | | Distress | ≥0.9 | 31.4 | SCL-90 | |
| | | | | | PTSD | ≥33 | 27.7 | IES-R | |
| Wong, Iran [98] | General population | Not reported | 55.8 | 1789 | Anxiety | ≥44 | 68 | STAI | 8 |
| Preis, US [69] | General population: Pregnant women | 29.19 (5.29) | 100 | 788 | Anxiety | ≥10 | 78.8 | GAD-7 | 8 |
| Wu, China [14] | General population: Pregnant women | 30 (27–32) | 100 | 4124 | Depression | ≥10 | 29.6 | EPDS | 8 |
| de Bruin, US [15] | General population | 48.56 (16.62) | 52 | 6666 | Anxiety | ≥3 | Not reported | PHQ-4 | 8 |
| | | | | | Depression | | | | |
| Kavčič, Slovenia [85] | General population | 36.4 (13.1) | 74.9 | 2722 | Stress | ≥17 | 54.4 | PSS | 7 |
| Zhou, China [94] | General population: Junior and Senior high school students and College students | 17.41 (2.7) | 57.7 | 11835 | Insomnia | >5 | 23.2 | PSQI | 7 |
| Zhu, China [89] | General population | 37.84 (7.69) | 55.5 | 922 | Distress | >160 | 18.3 | SCL-90 | 6 |
| Wang, China [51] | Patients: COVID-19 | 52.5 (14.3) | 50.2 | 484 | Insomnia | ≥8 | 42.8 | ISI-7 | 7 |
| Qi, China [87] | Patients: COVID-19 | 40.1 (10.1) | 58.1 | 41 | Anxiety/ | ≥50 | 26.8 | SAS/ | 8 |
| | | | | | Depression | ≥53 | 12.2 | SDS | |
| | | | | | PTSD | ≥4 | | PCL-C | |
| Wang, China [21] | General population | 32.32 (9.98) | 67.7 | 4827 | Anxiety | ≥10 | 53.3 | GAD-7 | 7 |
| | | | | | Depression | ≤13 | 48.3 | WHO-5 | |
| Zhou, China [65] | General population | Not reported | 67.8 | 2435 | Insomnia | ≥6 | Not reported | AIS | 6 |
| | | | | | Stress | ≥29 | | CPSS | |
| Tang, China [53] | General population | Not reported | 60 | 1160 | Anxiety | ≥5 | 70.8 | GAD-7 | 8 |
| | | | | | Depression | ≥15 | 26.5 | CES-D | |
| Verma, India [54] | General population | Not reported | 48.3 | 354 | Anxiety | >7 | 28 | DASS-21 | 6 |
| | | | | | Depression | >9 | 25.1 | | |
| | | | | | Stress | >14 | 11.6 | | |
| Gualano, Italy [55] | General population | 42 (23) | 65.6 | 1515 | Anxiety | ≥3 | 23.2 | GAD-2 | 8 |
| | | | | | Depression | ≥3 | 24.7 | PHQ-2 | |
| | | | | | Insomnia | ≥8 | 42.2 | ISI | |
| Kokou-Kpolou, France [59] | General population | 18–87 | 75.5 | 556 | Insomnia | ≥15 | 19.1 | ISI | 6 |
| Mechili, Albania [61] | General population | 18–85 | 84.6 | 1112 | Depression | ≥10 | 49.6 | PHQ-9 | 7 |
| Lin, China [63] | General population | 18–70 | 70 | 2446 | Anxiety | ≥40 | 78.3 | STAI | 8 |

*(Continued)*

**Table 1.** (*Continued*)

| Author, study location | Participant information | Age (mean, median or range) | Gender Female (%) | Sample size | Psychological distress | Cut-off points | Prevalence of psychological distress (%) | Psychometric instruments | Study quality |
|---|---|---|---|---|---|---|---|---|---|
| Ueda, Japan [66] | General population | Not reported | 50.4 | 2000 | Anxiety | ≥10 | 31.6 | GAD-7 | 7 |
| | | | | | Depression | ≥10 | 43.2 | PHQ-9 | |
| Naser, Jordan [67] | General population | Not reported | 59 | 4126 | Anxiety | ≥15 | 32.1 | GAD-7 | 7 |
| | | | | | Depression | ≥15 | 43.9 | PHQ-9 | |
| Li, UK [75] | General population | Not reported | Not reported | 15330 | Distress | ≥4 | 29.2 | GHQ-12 | 7 |
| Fekih-Romdhane, Tunisia [76] | General population | 29.2 (10.4) | 74 | 603 | PTSD | >33 | 33 | IES-R | 7 |
| Shi, China [78] | General population | 35.97 (8.22) | 52.1 | 56679 | Anxiety | ≥5 | 31.6 | GAD-7 | 8 |
| | | | | | Depression | ≥5 | 27.9 | PHQ-9 | |
| | | | | | Insomnia | ≥8 | 29.2 | ISI | |
| Qi, China [79] | General population: Adolescents | 11–20 | Not reported | 9554 | Anxiety | ≥5 | 19 | GAD-7 | 8 |
| Peng, China [82] | General population | 18–70 | 41.7 | 2237 | Depression | >50 | 6.21 | SDS | 8 |
| Fu, China [86] | General population | Not reported | 69.7 | 1242 | Anxiety | ≥5 | 27.5 | GAD-7 | 8 |
| | | | | | Depression | ≥5 | 29.3 | PHQ-9 | |
| | | | | | Insomnia | ≥5 | 30 | AIS | |
| Palgi, Israel [64] | General population | 46.21 (16.49) | 75.2 | 1059 | Anxiety | ≥10 | 19 | GAD-7 | 7 |
| | | | | | Depression | ≥10 | 14.4 | PHQ-9 | |
| Seyahi, Turkey [88] | Patients with rheumatic diseases, HCWs, Teachers/Academics | 16–81 | 69.2 | 2223 | Anxiety | ≥8 | 24.7 | HADS | 8 |
| | | | | | Depression | ≥8 | 45.3 | IES-R | |
| | | | | | PTS | ≥33 | 29.7 | | |
| Lee, China [97] | General population | Not reported | 75.8 | 3064 | Depression | ≥5 | Not reported | PHQ-9 | 7 |
| Duan, China [60] | General population: Children and Adolescents | 7–18 | 49.8 | 3613 | Depression | ≥19 | 22.28 | CDI | 7 |
| Karatzias, Ireland [72] | General population | Not reported | 51.5 | 1041 | PTSD | NA | 17.7 | ITQ | 8 |
| Liu, China [73] | General population: Pregnant women | 14–60 | 100 | 1947 | Anxiety | ≥50 | 17.2 | SAS | 8 |
| Yang, China [74] | General population | 36.3 (9.1) | 49.2 | 2410 | Insomnia | >5 | 14.9 | CPSQI | 8 |
| Hou, China [77] | General population: Senior high school students | Not reported | 38.6 | 859 | Anxiety | ≥10 | 54.5 | GAD-7 | 6 |
| | | | | | Depression | ≥8 | 71.5 | PHQ-9 | |
| Wang, China [80] | General population: College students | 21 (2.4) | 54.5 | 44447 | Anxiety | ≥50 | 7.7 | SAS | 8 |
| | | | | | Depression | ≥28 | 12.2 | CES-D | |
| Ma, China [81] | Patients: COVID-19 | 50.43 (13.12) | 51.9 | 770 | Depression | ≥5 | 43.1 | PHQ-9 | 8 |
| Domínguez-Salas, Spain [83] | General population | 40.26 (13.18) | 74 | 4615 | Distress | ≥3 | 72 | GHQ-12 | 8 |
| Huang, China [90] | General population | Not reported | 57.3 | 6261 | Anxiety | ≥50 | 13.5 | SAS | 8 |
| | | | | | Depression | ≥10 | 17.2 | PHQ-9 | |
| Liu, China [92] | Patients: COVID-19 | 55 (41–66) | 53 | 675 | Anxiety | ≥10 | 10.4 | GAD-7 | 8 |
| | | | | | Depression | ≥10 | 19 | PHQ-9 | |
| | | | | | PTSD | - | 12.4 | PCL-5 | |
| Ramasubramanian, India [95] | General population | Not reported | 64.2 | 2317 | Distress | ≥28 | 22.8 | CPDI | 6 |
| Ben-Ezra, China [50] | General population | 30.99 (6.82) | 53.5 | 1134 | Distress | ≥13 | 19.1 (severe) | K6 | 7 |

(*Continued*)

**Table 1.** (Continued)

| Author, study location | Participant information | Age (mean, median or range) | Gender Female (%) | Sample size | Psychological distress | Cut-off points | Prevalence of psychological distress (%) | Psychometric instruments | Study quality |
|---|---|---|---|---|---|---|---|---|---|
| Mosli, Saudi Arabia [99] | Patients: Inflammatory bowel disease | Not reported | 47.5 | 1156 | Anxiety | ≥11 | 63.4 | HADS-A | 8 |
| | | | | | Depression | ≥11 | 30.1 | HADS-D | |

**Abbreviations:** DASS-21, Depression, Anxiety, and Stress Scale; K10, Kessler-10; GAD, Generalised Anxiety Disorder; PHQ, Patient Health Questionnaire; HAM-A, Hamilton Anxiety Rating Scale; HAM-D, Hamilton Depression Rating Scale; FCV-19S, Fear of COVID-19 Scale; CES-D, Center for Epidemiological Studies Depression scale; PSQI, Pittsburgh Sleep Quality Index; ISI, Insomnia Severity Index; HADS, Hospital Anxiety and Depression Scale; SCL-90, Symptom Checklist-90; SCL-90-R, Symptom Checklist-90-Revised; WHO-5, World Health Organisation Five Well Being Index; SCARED, Screen for Child Anxiety Related Emotional Disorders; CDI-S, Children's Depression Inventory-Short Form; K6, Kessler-6; PSS, Perceived Stress Scale; SSS, Somatic Self-rating Scale; SAS, Self-Rating Anxiety Scale; SDS, Self-Rating Depression Scale; GHQ, General Health Questionnaire; DSRS-C, Depression Self-rating Scale for Children; PTSS, Post-Traumatic Stress Symptoms; PTSD, Post-Traumatic Stress Symptoms Disorders; DSM-5, Diagnostic and Statistical Manual of Mental Disorders, Fifth Edition; BAI, Becks Anxiety Inventory; BDI, Becks Depression Inventory; SDQ EPS, Strengths and Difficulties Questionnaire Emotional Problems Scale; PCL-C, PTSD Checklist—Civilian Version; PCL-5, PTSD Checklist for DSM-5; CPDI, COVID-19 Peritraumatic Distress Index; PSS-10-C, Perceived Stress Scale modified for COVID-19; STAI, State-Trait Anxiety Inventory; IES-R, Impact of Event Scale-Revised; EPDS, Edinburgh Postnatal Depression Scale; EQ-5D, EuroQol—Five Dimensions; AIS, Athens Insomnia Scale; CPSS, Chinese version of the Perceived Stress Scale; PTS, Posttraumatic stress; CDI, Child Depression Inventory; ITQ, International Trauma Questionnaire; CPSQI, Chinese version of Pittsburgh Sleep Quality Index; HCWs, Healthcare Workers; NHCWs, Non-Healthcare Workers.

Included studies collected information on sociodemographic data, history of physical and mental diseases, media exposure, family and social support, positive coping strategies, and psychological distress of the COVID-19 pandemic. Anxiety [8–13, 15–22, 26, 47, 49, 53–57, 63, 64, 66, 67, 69, 73, 77–80, 84, 86–88, 90, 92, 96, 98, 99] and depression [8–18, 20–22, 26, 49, 52–57, 60, 61, 64, 66, 67, 70, 77, 78, 80–82, 84, 86–88, 90, 92, 96, 97, 99] were the most common indicators of psychological distress reported by included studies. The overall prevalence of anxiety was 33% (95% CI: 28%-39%; $I^2$ = 99.9%) among the predominantly general population, and the prevalence of depression was 30% (95% CI: 26%-36%; $I^2$ = 99.8%). In addition, nine studies (13.4%) reported distress [48, 50, 62, 75, 83, 89, 91, 93, 95], six studies (8.96%) reported stress [10, 54, 65, 71, 84, 85], 12 studies (17.9%) reported insomnia [12, 51, 52, 55, 57–59, 65,

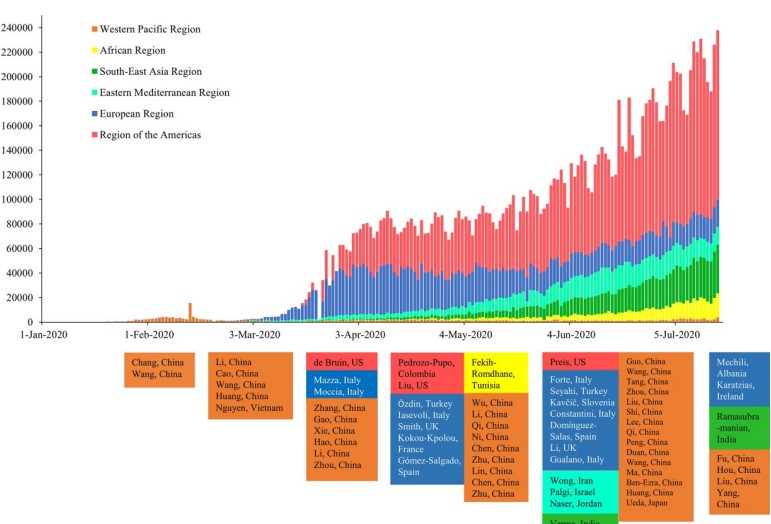

**Fig 2. Daily new cases of COVID-19 and publication dates of included papers by by six World Health Organization regions and time.**

74, 78, 86, 94], and nine studies (13.4%) reported post-traumatic stress disorder/symptoms (PTSD/PTSS) [11, 52, 56, 72, 76, 77, 87, 88, 92].

A variety of validated psychometric instruments were used to measure symptoms of anxiety and depression. The most often used tool was the Generalized Anxiety Disorder-7 for anxiety, and the Patient Health Questionnaire-9 for depression. In addition, common tools for other indicators of psychological distress included the General Health Questionnaire-12 (for distress), the Depression, Anxiety, and Stress Scale-21 (for stress), the Insomnia Severity Index (for insomnia) and the Impact of Event Scale-Revised (for PTSD/PTSS) (Table 1).

## Women

A total of 50 studies with 82 data points reported the association between women and higher odds of psychological distress, as most studies reported more than one indicator of psychological distress. The pooled OR of women versus men was 1.48 (95% CI: 1.29–1.71; $I^2$ = 90.8%) for anxiety and 1.16 (1.07–1.26; $I^2$ = 75.0%) for depression (Fig 3). The significant OR persisted for secondary outcomes of distress (1.83 [1.63–2.06]; $I^2$ = 50.8%), and was borderline significant for insomnia (1.20 [0.98–1.47]; $I^2$ = 91.7%) and PTSD/PTSS (1.82 [0.97–3.40]; $I^2$ = 93.5%) (S1 Fig in S1 File).

## Younger age

A total of 37 studies reported the association between age and higher odds of psychological distress with 62 data points. Younger age (majority <35 years) versus older age (≥35 years) was associated with higher odds of primary outcomes of psychological distress. The pooled OR of younger versus older age was 1.20 (1.13–1.26; $I^2$ = 91.7%) for anxiety, and 1.13 (1.08–1.18; $I^2$ = 95.1%) for depression (Fig 4). In terms of secondary outcomes of psychological distress, the OR was significant for stress (1.08 [1.03–1.14]; $I^2$ = 91.1%), and borderline significant for distress (1.02 [0.98–1.05]; $I^2$ = 97.1%) (S2 Fig in S1 File).

## Lower SES

Lower SES strata was associated with higher odds of psychological distress (Figs 5–7 & S3 Fig in S1 File). The pooled OR of lower versus higher education from 30 studies (48 data points) was 1.21 (1.05–1.40; $I^2$ = 86.1%) for anxiety and 1.15 (1.03–1.29; $I^2$ = 82.0%) for depression; the significant association was also observed for stress (1.15 [1.03–1.29]; $I^2$ = 9.0%) (Fig 5 & S3 Fig in S1 File). The pooled OR of lower versus higher income from 15 studies (26 data points) was 1.45 (1.24–1.69; $I^2$ = 82.3%) for anxiety and 1.56 (1.26–1.92; $I^2$ = 85.4%) for depression; the significant OR persisted for stress (1.27 [1.20–1.34]; $I^2$ = 0%) (Fig 6 & S3 Fig in S1 File). Current employment (yes vs. no) was associated with lower odds of psychological distress. The pooled OR from 11 studies (21 data points) was 0.89 (0.78–1.02; $I^2$ = 26.6%) for anxiety and 0.76 (0.61–0.95; $I^2$ = 63.8%) for depression (Fig 7).

## Rural dwelling

Nine studies with 14 data points compared risk of living in rural versus urban areas. A significant association has been observed with anxiety. The pooled OR was 1.13 (1.00–1.29; $I^2$ = 82.9%) for anxiety and 0.98 (0.85–1.12; $I^2$ = 81.6%) for depression (Fig 8).

## Higher COVID-19 infection risk

Indicators of higher COVID-19 infection risk were consistently associated with higher odds of psychological distress (S4 and S5 Figs in S1 File). The pooled OR of suspected/confirmed

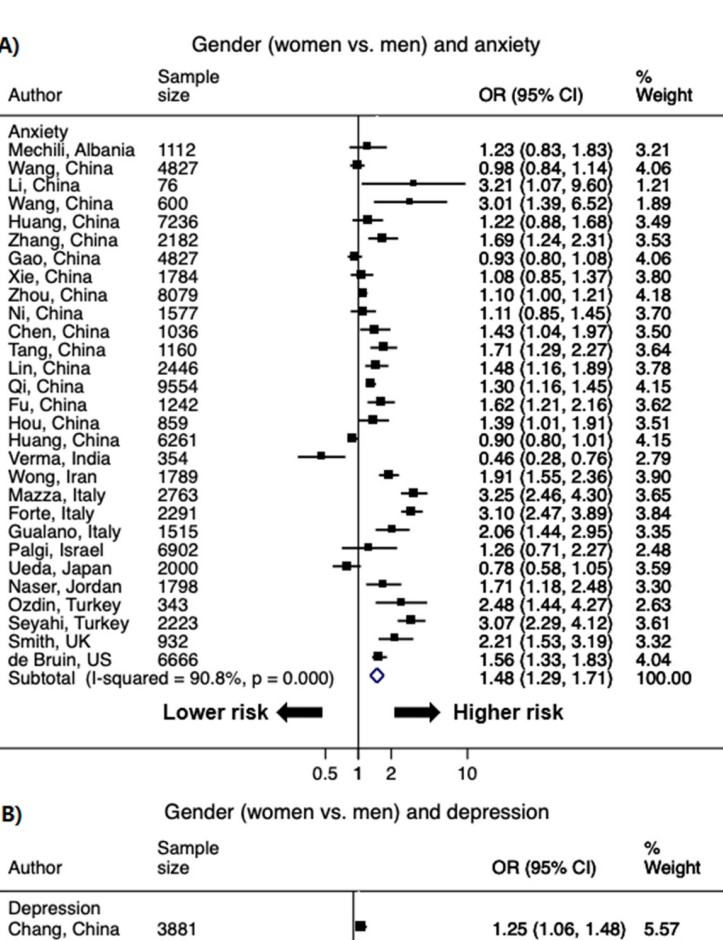

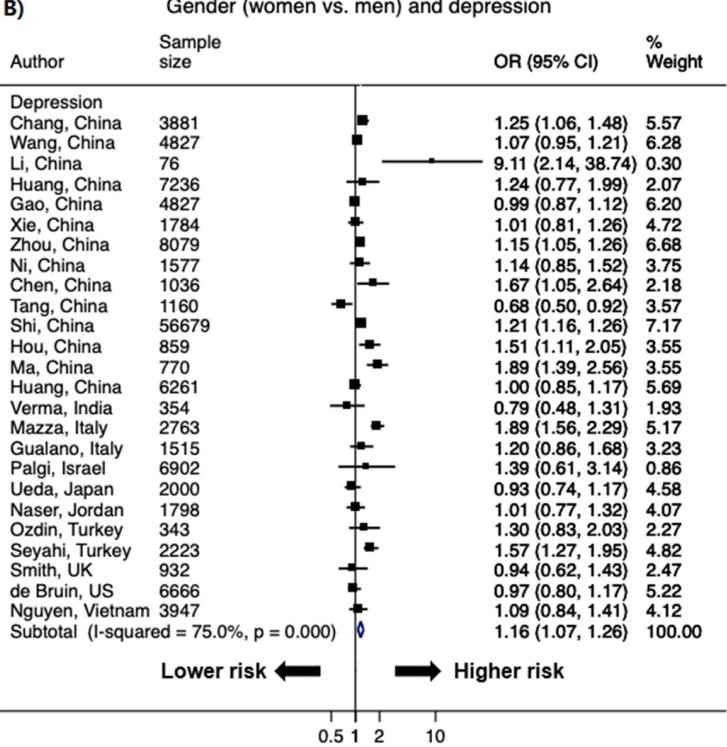

**Fig 3.** Forest plot of the association between gender (women versus men) and A) anxiety and B) depression. The size of the data markers indicates the weight of the study, which is the inverse variance of the effect estimate. The diamond data markers indicate the pooled ORs.

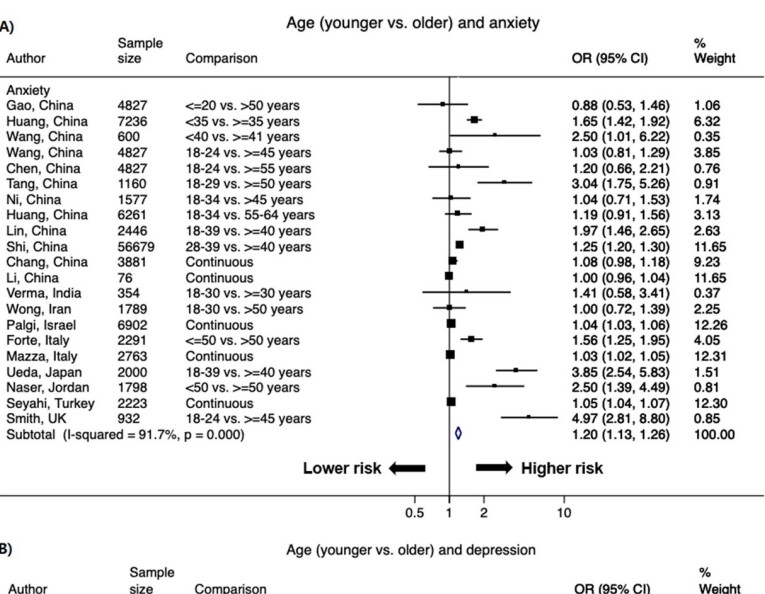

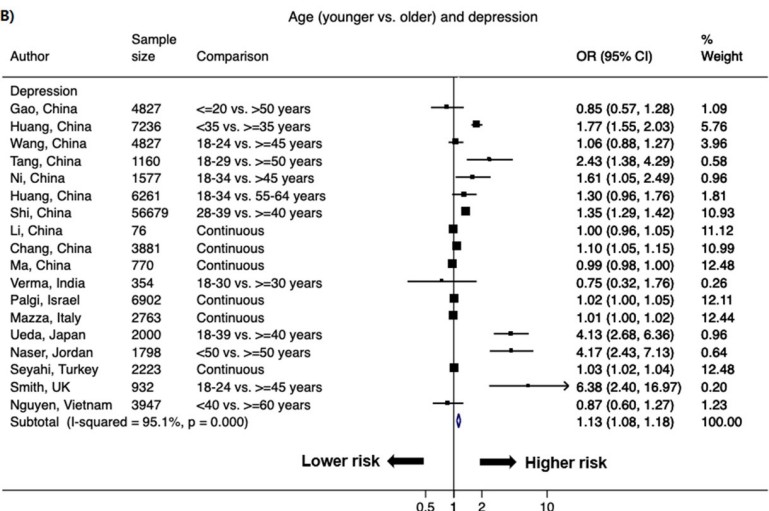

**Fig 4.** Forest plot of the association between age (younger vs. older) and A) anxiety and B) depression. The size of the data markers indicates the weight of the study, which is the inverse variance of the effect estimate. The diamond data markers indicate the pooled ORs.

COVID-19 cases (16 studies and 33 data points) was 1.70 (1.41–2.06; $I^2$ = 79.5%) for anxiety, 1.84 (1.39–2.43; $I^2$ = 73.5%) for depression, 1.28 (1.17–1.40; $I^2$ = 0%) for distress, 2.19 (1.56–3.08; $I^2$ = 59.0%) for insomnia, and 1.27 (1.10–1.47; $I^2$ = 0%) for PTSD/PTSS. The pooled OR of living in the hard-hit area (12 studies and 20 data points) was 1.57 (1.36–1.81; $I^2$ = 73.9%) for anxiety and 1.33 (1.16–1.53; $I^2$ = 69.1%) for depression. The pooled OR of having pre-existing physical conditions or worse health (17 studies and 29 data points) was 1.48 (1.21–1.81; $I^2$ = 65.2%) for anxiety, 1.42 (1.12–1.80; $I^2$ = 89.0%) for depression, 1.21 (1.12–1.31; $I^2$ = 0%) for stress, and 1.89 (1.30–2.73; $I^2$ = 86.6%) for insomnia. The OR of having mental health conditions (eight studies and 15 data points) was 1.82 (1.34–2.48; $I^2$ = 70.8%) for anxiety, 1.75 (0.98–3.14; $I^2$ = 93.5%) for depression, and 1.42 (1.11–1.82; $I^2$ = 59.7%) for insomnia.

## Other factors

Longer media exposure (ten studies and 20 data points) was associated with higher odds of anxiety (1.57 [1.16–2.13]; $I^2$ = 94.5%), depression (1.34 [1.12–1.60]; $I^2$ = 86.2%), insomnia

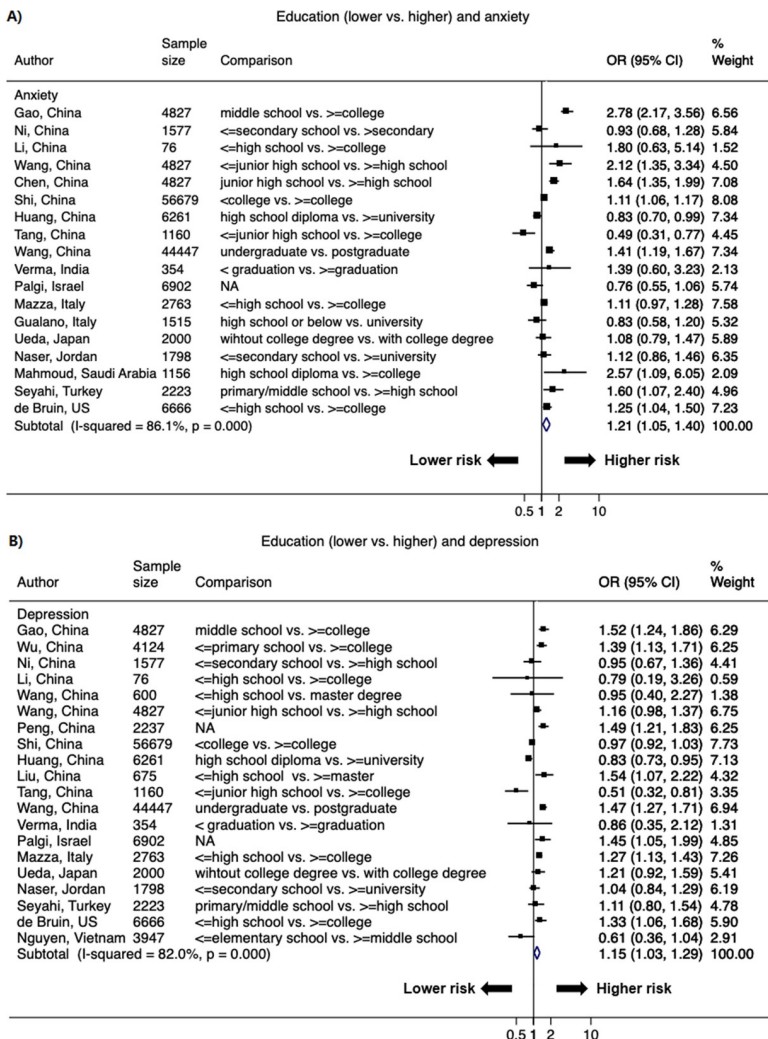

**Fig 5.** Forest plot of associations between education (lower vs. higher) and A) anxiety and B) depression. The size of the data markers indicates the weight of the study, which is the inverse variance of the effect estimate. The diamond data markers indicate the pooled ORs.

(1.04 [1.00–1.08]; $I^2$ = 0%), and PTSD/PTSS (1.48 [1.23–1.78]; $I^2$ = 0%) (S6 and S7 Figs in S1 File). In addition, social/family support and physical activity were inversely associated with higher odds of anxiety and depression (S6 Fig in S1 File). The pooled OR of social/family support (6 studies and 9 data points) was 0.68 (0.58–0.79; $I^2$ = 0%) for anxiety and 0.47 (0.40–0.56; $I^2$ = 0%) for depression. The pooled OR of longer physical activity (7 studies and 11 data points) was 0.71 (0.58–0.88; $I^2$ = 52.3%) for anxiety and 0.69 (0.50–0.94; $I^2$ = 84.8%) for depression.

## Assessment of heterogeneity

We observed a substantial heterogeneity for most of the associations between studies ($I^2$ range: 52.3%-95.1%), except for the association between current employment and anxiety ($I^2$: 26.6%; $P$ = 0.22) and the associations of family/social support with both anxiety ($I^2$: 0%; $P$ = 0.58) and depression ($I^2$: 0%; $P$ = 0.58). We further conducted stratified analyses to explore the heterogeneity. We found that the OR observed in the overall population was largely consistent across

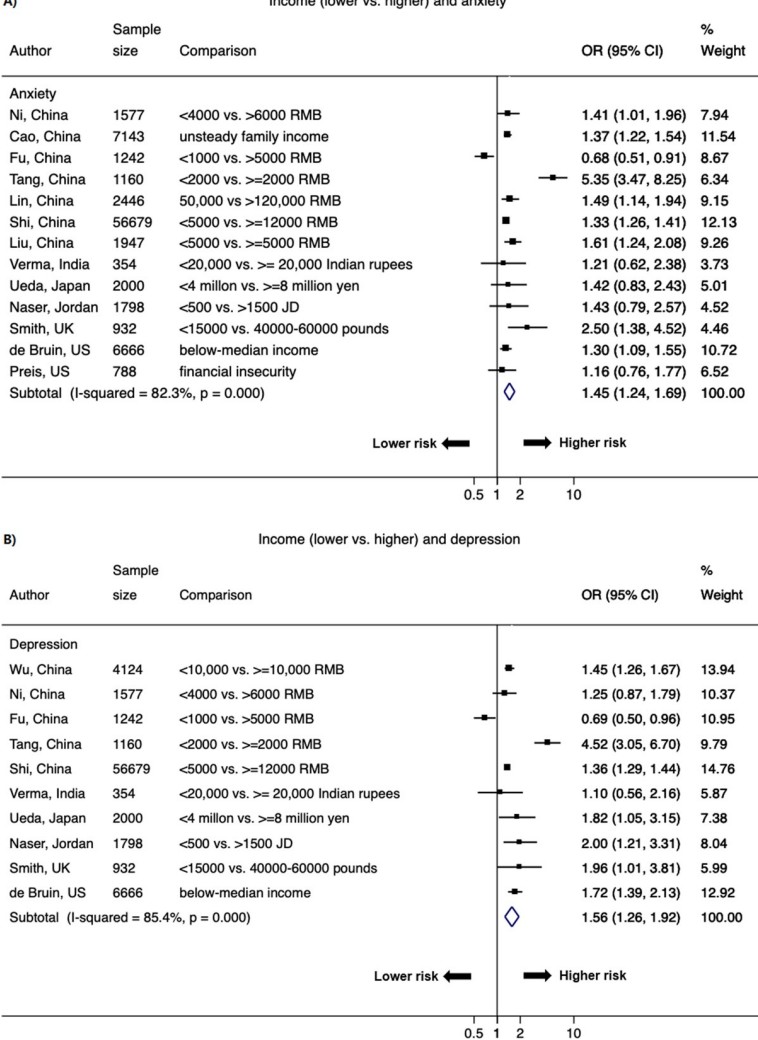

**Fig 6.** Forest plot of associations between income (lower vs. higher) and A) anxiety and B) depression. The size of the data markers indicates the weight of the study, which is the inverse variance of the effect estimate. The diamond data markers indicate the pooled ORs.

all subgroups by locations or instruments/cut-off points containing three or more studies, albeit the 95% CI became wider for subgroups with fewer studies and did not achieve statistical significance for subgroups with very few studies (S2 and S3 Tables in S1 File). Moreover, except for the association of anxiety with gender (higher odds in studies in Asian than in Europe, *P* for meta-regression = 0.037), no statistically significant differences were found across subgroups of study locations for other factors (all *P*-values from meta-regression ≥0.09), indicating that geographic differences were less likely to influence the observed associations in the current meta-analysis (S2 Table in S1 File). For psychometric instruments, only studies assessing the association between gender and anxiety offered enough power to have three subgroups (containing three or more studies), while other factors only had one subgroup. The meta-regression analysis suggested no statistical differences across subgroups of studies using different instruments or cut-off points (*P*-value from meta-regression = 0.66) (S3 Table in S1 File).

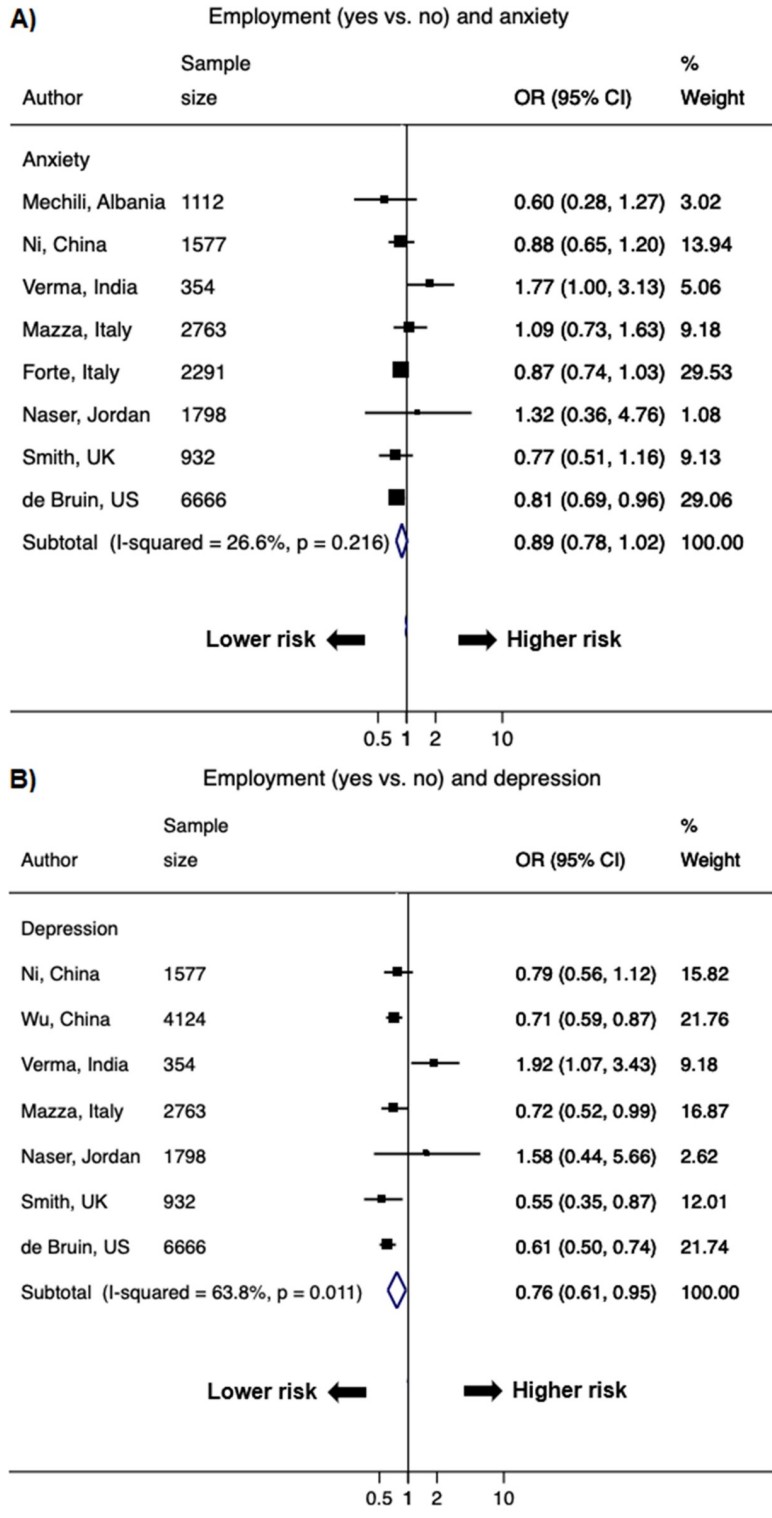

**Fig 7.** Forest plot of associations between employment (yes vs. no) and A) anxiety and B) depression. The size of the data markers indicates the weight of the study, which is the inverse variance of the effect estimate. The diamond data markers indicate the pooled ORs.

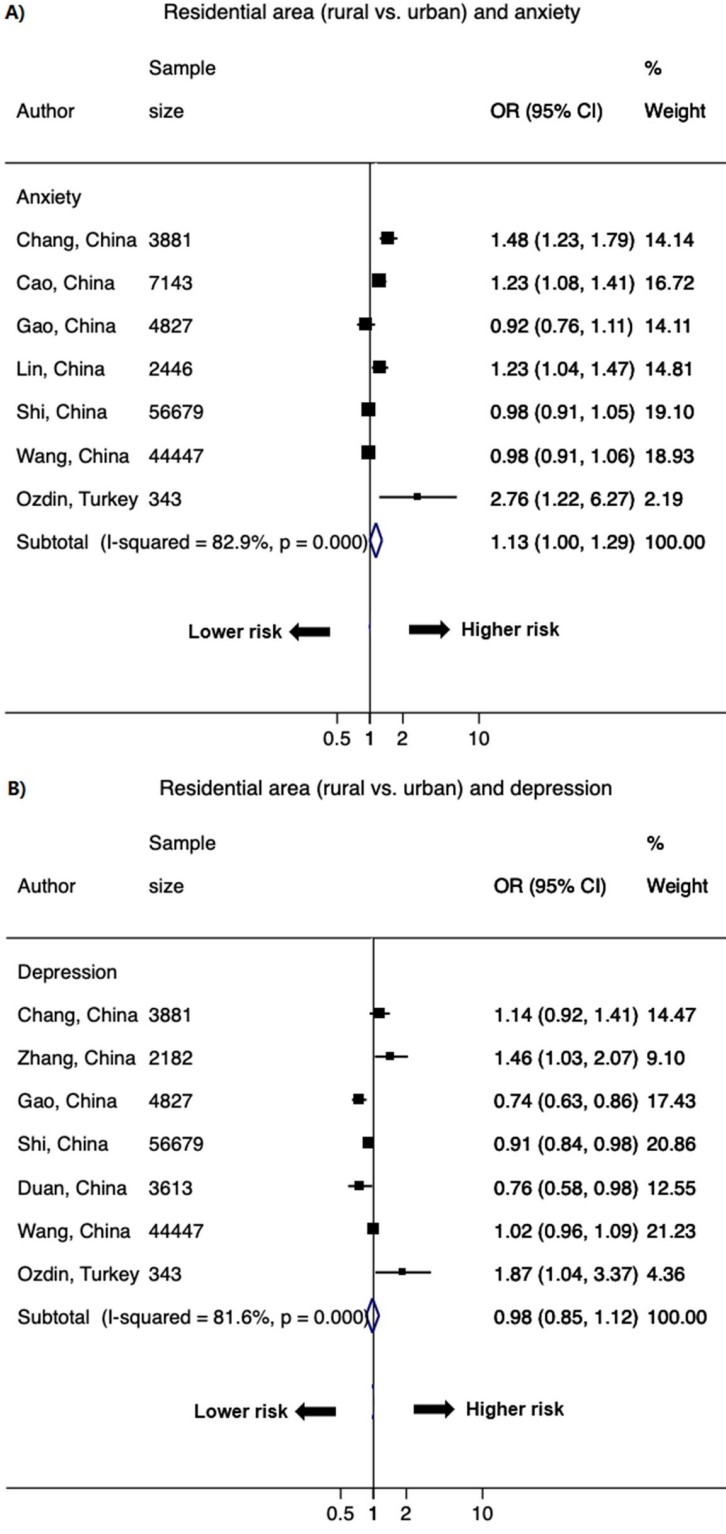

**Fig 8.** Forest plot of the association between residential area (rural vs. urban) and A) anxiety and B) depression. The size of the data markers indicates the weight of the study, which is the inverse variance of the effect estimate. The diamond data markers indicate the pooled ORs.

## Sensitivity analyses

We further repeated the aforementioned analyses excluding 11 studies containing high-risk and vulnerable populations, or a small subset of healthcare workers. We found that all significant associations remained essentially the same across all factors, indicating that the heterogeneity in populations did not impact our results (S4 Table in S1 File).

## Assessment of publication bias

Studies on anxiety and depression with gender, age, and SES strata (lower education, lower income) had enough power to test for publication bias with funnel plots ($n \geq$ten). Visual inspection of funnel plots revealed asymmetry for age with anxiety and depression (S8 Fig in S1 File), and the Egger's test was statistically significant ($P \leq 0.01$). After applying the trim-and-fill method, the pooled RR was 1.08 (1.02–1.15) for anxiety and 1.06 (1.01–1.11) for depression, suggesting that the potential publication bias did not affect the significant associations of age with anxiety and depression. No significant asymmetry was observed for gender and SES strata with anxiety and depression, indicating the unlikelihood of publication bias for these factors (S8 Fig in S1 File).

## Attributable risk

Two studies (one from China and one from Vietnam) reported the prevalence of depression among patients with and without COVID-19 [70, 81]; the prevalence of depression among patients with COVID-19 was 51.6% in China and 13.6% in Vietnam, and the prevalence of depression among patients without COVID-19 was 41.9% in China and 7.04% in Vietnam. The attributable risk of depression due to the COVID-19 pandemic was 9.70% in China and 9.52% in Vietnam (S5 Table in S1 File).

## Discussion

Using data from 68 cross-sectional studies of 288,830 participants from 19 countries, our meta-analysis found that one in three adults in the predominantly general population have anxiety or depression. Women, younger adults, individuals of lower SES strata (lower education, lower income, unemployment), residing in rural areas, and those with or at high risk of COVID-19 infection (suspected/confirmed cases, living in the hard-hit area, having history of chronic conditions or mental conditions) were associated with higher odds of psychological distress. Our results underscore the importance of allocating mental health resources and evaluation of approaches including risk stratification and targeted intervention among individuals at high risk of psychological distress due to the COVID-19 pandemic. Our results also show that improving family and social support and positive coping strategies are associated with reduced risk of psychological distress.

Consistent with our findings on the prevalence of anxiety (33% [28%-39%]) and depression (30% [26%-36%]), a recent meta-analysis that solely focused on the prevalence of psychological distress during the COVID-19 pandemic also found that about one in three adults in the general population had anxiety (33% [28%-38%]) or depression (28% [23%-32%]), respectively [6]. We also observed similar attributable risk of psychological distress due to COVID-19 (around 10%) at the start of the outbreak in February 2020 from two studies (from China [81] and Vietnam [70]) with available data, albeit the prevalence of depression among patients with and without COVID-19 was higher in China (51.6% and 41.9%) compared to that in Vietnam (13.6% and 7.04%) possibility due to the higher infection rate in China (0.56 per 10,000 people [103] vs. 0.0017 per 10,000 people in Vietnam [104]). The prevalence of depression among

COVID-19 patients observed in the Chinese study was similar to that reported in a recent meta-analysis among COVID-19 patients (45% [95% CI: 37%-54%]) [36]. As the impact of COVID-19 has become substantially wider globally, the attributable risk is likely to have increased as the pandemic evolves.

Several factors identified for higher risk of psychological distress during previous infectious outbreaks (e.g. severe acute respiratory syndrome [SARS]), such as women [105], individuals of lower SES strata (e.g. lower education levels, lower income levels) [106], and those with higher risk of disease exposure [107], corroborated our findings for the current COVID-19 pandemic. Older age has been found to be associated with higher risk of COVID-19 infection [108]; however, it is interesting to observe that younger people (mainly <35 years) had higher odds of psychological distress during the COVID-19 pandemic. Although we observed an asymmetric funnel plot of age suggesting potential publication bias, the association of age with anxiety and depression remained statistically significant after applying the trim-and-fill method, indicating that the potential publication bias is unlikely to affect the observed associations of age with anxiety and depression. Although the underlying mechanisms are not clear yet, some studies suggested that the higher odds of psychological distress in younger people could be due to their greater access to COVID-19 information through media [10, 109]. In corroboration, longer media exposure was associated with higher odds of psychological distress in the current study (OR: 1.57 [95% CI: 1.16–2.13] for anxiety; 1.34 [1.12–1.60] for depression). Furthermore, with abrupt closure of educational institutions and workplaces, younger adults might be more concerned about their future prospects [60, 79]. In addition, younger people were less likely to have experienced previous infectious outbreaks (e.g. SARS) compared to their older counterparts. A study conducted in Hong Kong has found that not experiencing the SARS outbreak in 2003 is associated with higher risk of psychological distress of COVID-19 and suggested that the first experience of an infectious disease outbreak is an incredibly stressful event [110].

Our observed positive association between female and higher odds of psychological distress was consistent with results from the Global Burden Disease of Study 2015 that anxiety and depression were more common in women (4.6% and 5.1%) than men (2.6% and 3.6%) [7]. The reasons for the gender disparities are largely unknown. Although differences in physical strength, variations in ovarian hormone levels and decreases in estrogen may play some role [110], the lower social status of women and less preferential access to healthcare compared to men could potentially be responsible for the exaggerated adverse psychosocial impact on women [111]. Of note, the rates of suiside and self-harm are already high in women globally [112, 113], and the rates were expected to increase even after the COVID-19 pandemic [114]. Thus, outreach programs for mental health services must target women proactively.

Previous studies conducted during non-COVID period have found that people living in urban areas are at higher risk of psychological distress, possibly due to higher rates of pollution, specific urban designs (less access to green area [115], tall buildings, population density that may be perceived as oppressive), and more physical threats (accidents, violence) [116–120]. However, we found higher odds of anxiety among rural versus urban dwellers in our analysis, albeit the association with depression was not significant. As rural areas during the COVID-19 pandemic may be a reflection of the poorer healthcare infrastructure, economy, sanitation, and educational resources [19], the observed rural-urban gradient has important public health implication of ensuring equitable healthcare resources in rural and resource-restraint areas during the COVID-19 pandemic, and further studies are warranted to investigate the rural-depression association. Nevertheless, according to the United Nations, in many countries, the reverse might be true, whereby urban areas may have poorer environment and infrastructure for the prevention of COVID-19 [121]. In the current meta-analysis, we had

relatively small sample size for the association between residential area and psychological distress (nine studies for anxiety and depression), and further studies with larger sample sizes and higher statistical power are warranted to examine this association. Furthermore, we found that lower SES, in particular lower income and lower education, both, were associated with higher odds of anxiety, depression and stress during the pandemic. Although the reasons and pathways triggering the psychological distress were not explored, it is possible that the anticipated burden of potential treatment expenses as well as loss of income opportunities related to pandemic affect those already living with limited means. The association with lower education probably reflects low health literacy, low perceptions of personal risk, and lack of awareness regarding coping mechanisms [122]. Clearly, these vulnerable populations seem to be at the greatest need for preventative mental health services. Therefore, our findings highlight the importance of equitable healthcare delivery solutions especially in socioeconomically disadvantaged and resource-restraint areas for addressing the high burden of COVID-19 related psychological distress.

Our results had important clinical and public health implications. First, the identified risk factors of higher odds of psychological distress of COVID-19 could be used to identify and recognize populations with higher risk of psychological distress. According to the NICE's "stepped-care" framework, low-intensity psychosocial interventions (social/family support, education programs, individual guided or computerized self-help cognitive behavioral therapy, physical activity programs) would be initiated for people with milder depression, whereas high-intensity interventions (formal psychological therapies by trained therapists) will be initiated for people with severe symptoms [123]. In addition, task-shifting approaches with trained lay counsellor-delivered brief psychological treatment has been shown to be effective in the treatment of depressive mental disorders in resource-challenged settings [124]. Therefore, a variety of approaches coupled with telehealth need to be considered to urgently target the high-risk populations identified in our study–women, younger adults, individuals of lower SES strata, and those with or at high risk of COVID-19 infection.

Previous meta-analyses among general population only reported the prevalence of psychological distress [6, 32–37]. In comparison, the primary objective of the current meta-analysis was to determine the factors associated with psychological distress using quantitative assessment during the COVID-19 pandemic in the predominantly general population. We are not aware of other meta-analyses assessing factors of psychological distress in the general population. Prior to the literature search, we registered our pre-defined study protocol with the National Institute for Health Research International prospective register of systematic reviews (PROSPERO, #CRD42020186735) [38]. We followed both PRISMA [30] and MOOSE [31] guidelines to conduct the current meta-analysis, and performed quality assessment of included studies using the validated instrument. The included studies covered all six WHO continents geographically and consisted of both low-and middle-income countries and high-income countries, which ensured great generalizability of our results. Since the majority of publications were from China, where the first infected case with COVID-19 was identified, we included papers published both in English and Chinese to maximize the search results.

However, some limitations merit consideration. First, we only included peer-reviewed publications in the current meta-analysis, and we did not explore potentially relevant grey literature. Nevertheless, this is to ensure the quality of the included publications. Second, we included a predominantly general population for the current meta-analysis, with a few studies among high-risk and vulnerable patients, and a small subset of healthcare workers among studies using random sampling techniques, which may bring potential heterogeneity to our results. Nevertheless, we conducted sensitivity analyses and repeated all analyses among studies only containing general populations, and found the same significant results across all

factors, indicating that the heterogeneity in study population did not impact our results. Third, psychological distress was measured using self-reported questionnaires, which may have brought bias to an overestimation or underestimation of the prevalence of psychological distress. However, for associations between factors and psychological distress, self-reported bias would lead to non-differential misclassifications and result in an underestimation of the true effect size of factors associated with psychological distress. Therefore, the significant associations between factors and psychological distress observed in the current study are unlikely to be impacted by self-reported questionnaires. Fourth, instruments to measure indicators of psychological distress (e.g. anxiety, depression) were not identical in all studies. However, we only included studies that used standardized and validated instruments to measure psychological distress, which mitigates the possibility of systematic bias to our best extent. Furthermore, the data points vary for the different outcomes, and we observed a substantial heterogeneity for most of the associations. However, we have included all published studies reporting the association for the outcomes of interest during the pandemic period covered during our study. In addition, we conducted meta-regression and stratified analyses by study locations and different instruments and cut-off points, and observed largely non-significant P-values for meta-regression analyses and consistent point estimates of OR across subgroups, suggesting no substantial statistical differences across subgroups. Nevertheless, the sample sizes and numbers of subgroups were small for the stratified analyses, thus the 95% CI became wider and did not achieve statistical significance for subgroups with very few studies; future studies with larger sample sizes are warranted to validate our findings.

In conclusion, our meta-analysis of 68 studies found that one in three adults in the predominantly general population have anxiety or depression. Women, younger adults, individuals residing in rural areas, of lower SES strata, those with or at high risk of COVID-19 infection, and longer media exposure were associated with higher odds of psychological distress. Our findings highlight the urgent need for offering mental health services and interventions to target high-risk populations to reduce socioeconomic and gender disparities of psychological distress during the COVID-19 pandemic globally.

## Supporting information

**S1 Checklist. PRISMA 2009 checklist.**
(DOC)

**S1 File.**
(DOCX)

**S1 Dataset.**
(XLSX)

## Acknowledgments

We thank the following study investigators for clarifying inquiry of their papers: Dr. Emre Umucu, PhD, from The University of Texas at El Paso, USA; Dr. Yun Li, MD, from the Mental Health Center of Shantou University, China; and Dr. Feten Fekih-Romdhane, MD, from the University Tunis El Manar, Tunisia.

## Author Contributions

**Conceptualization:** Tazeen H. Jafar.

**Data curation:** Tazeen H. Jafar.

**Formal analysis:** Yeli Wang, Tazeen H. Jafar.

**Investigation:** Tazeen H. Jafar.

**Methodology:** Yeli Wang, Monica Palanichamy Kala, Tazeen H. Jafar.

**Project administration:** Tazeen H. Jafar.

**Resources:** Tazeen H. Jafar.

**Software:** Yeli Wang.

**Supervision:** Tazeen H. Jafar.

**Validation:** Monica Palanichamy Kala, Tazeen H. Jafar.

**Visualization:** Tazeen H. Jafar.

**Writing – original draft:** Yeli Wang, Monica Palanichamy Kala, Tazeen H. Jafar.

**Writing – review & editing:** Yeli Wang, Monica Palanichamy Kala, Tazeen H. Jafar.

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
