## [Decision Letter · Decision Letter 0]

29 Oct 2020

PONE-D-20-26693

Factors associated with psychosocial disorders during the coronavirus disease 2019 (COVID-19) pandemic on the general population: a systematic review and meta-analysis

PLOS ONE

Dear Dr. Jafar,

Thank you for submitting your manuscript to PLOS ONE. After careful consideration, we feel that it has merit but does not fully meet PLOS ONE’s publication criteria as it currently stands. Therefore, we invite you to submit a revised version of the manuscript that addresses the points raised during the review process.

Please see Additional Editor Comments.

We look forward to receiving your revised manuscript.

Kind regards,

Michio Murakami

Academic Editor

PLOS ONE

Journal Requirements:

2. Please clearly state your exclusion criteria.

3.Please include the funnel plot assessing publication bias in the Results section.

4. Please summarize results for the heterogeneity tests and indicate whether heterogeneity was high or low across studies. Also, what were the implications of these results on the feasibility of undertaking a meta-analysis?

5.We note that you have indicated that data from this study are available upon request. PLOS only allows data to be available upon request if there are legal or ethical restrictions on sharing data publicly. For information on unacceptable data access restrictions, please see http://journals.plos.org/plosone/s/data-availability#loc-unacceptable-data-access-restrictions.

6. We note that [Figure(s) 2] in your submission contain [map/satellite] images which may be copyrighted. All PLOS content is published under the Creative Commons Attribution License (CC BY 4.0), which means that the manuscript, images, and Supporting Information files will be freely available online, and any third party is permitted to access, download, copy, distribute, and use these materials in any way, even commercially, with proper attribution. For these reasons, we cannot publish previously copyrighted maps or satellite images created using proprietary data, such as Google software (Google Maps, Street View, and Earth). For more information, see our copyright guidelines: http://journals.plos.org/plosone/s/licenses-and-copyright.

1.    You may seek permission from the original copyright holder of Figure(s) [2] to publish the content specifically under the CC BY 4.0 license. 

Additional Editor Comments (if provided):

1. As pointed out by reviewer 1, if there are previous studies related to systematic reviews and meta-analysis, the statement in L362-364 needs to be revised.

2. Please add the cut-off points applied to the indicators in each study in Table 1.

3. The various indicators and cut-off points were analyzed together in this study. The strength of associations between mental health outcomes and factors can depend on the types of indicators and cut-off points. In discussing the strength of the association with each factor, the authors need to add further analyses to examine whether there are differences across indicators and cut-off points. If necessary, please perform a stratified analysis for each of the same indicators with the same cut-off points to deepen the discussion.

Reviewers' comments:

Reviewer's Responses to Questions

**Comments to the Author**

1. Is the manuscript technically sound, and do the data support the conclusions?

Reviewer #1: No

Reviewer #2: Yes

2. Has the statistical analysis been performed appropriately and rigorously? 

Reviewer #1: No

Reviewer #2: Yes

3. Have the authors made all data underlying the findings in their manuscript fully available?

Reviewer #1: Yes

Reviewer #2: Yes

4. Is the manuscript presented in an intelligible fashion and written in standard English?

Reviewer #1: Yes

Reviewer #2: Yes

5. Review Comments to the Author

Reviewer #1: This paper is a meta-analytic review of the factors associated with "psychosocial disorders" in the context of the COVID-19 pandemic.

There are several major problems with this paper in its current form:

1. The term "psychosocial disorders" is conceptually flawed. A "psychological disorder" (synonym: "mental disorder") is a medical-psychological term referring to a specific set of symptoms associated with mental distress and dysfunction. "Social disorder" is a general term referring to disturbances in social order such as civil unrest, famine or violence. To conflate the two and invent a new term ("psychosocial disorders") is erroneous.

2. The paper itself has only examined psychological disorders (as mentioned in the Abstract, the primary outcome measures are "anxiety" and "depression") and not social disorders such as poverty, job loss, civil unrest or lack of access to healthcare. Hence, the paper may be best given a new title using the term "psychological disorder" or "mental disorder". Yet even these terms are not without their problems in the context of this paper, as discussed immediately below.

3. The diagnosis of a psychological disorder is a formal process requiring the application of operationalized diagnostic criteria (such as the DSM-5) by a trained mental health care or allied professional. Though there are brief instruments, such as checklists or self-rated questionnaires, that can aid in this process, these are only screening instruments, and anyone screening positive on such an instrument would still need to have their diagnosis confirmed using the standard criteria mentioned above. The vast majority (if not all) of the studies included in this paper have only made use of screening instruments. Hence, strictly speaking, these studies cannot be used to estimate the frequency of "psychological disorder"; they can only assess the frequency of "symptoms of psychological disorder" (such as "depressive symptoms" or "anxiety symptoms"), or - to use a broad but still acceptable term - "psychological distress". Hence, "psychological distress" or "symptoms of X/Y/Z" would be an ideal term to use in the title and text of this article.

4. To my knowledge, there are already no less than three meta-analytic papers addressing the exact question that this paper examines - Deng et al. 2020, Bueno-Notivol et al. 2020, and Krishnamoorthy et al. 2020. This is not counting three meta-analyses exclusive to healthcare workers and other related papers. Hence, the need for another paper of this kind is unclear, especially since the authors of this paper have not acknowledged even one of these earlier meta-analyses in their own work. Unless the current paper represents a significant improvement or advance over these papers, either chronologically or in terms of methodology or quality, I see little justification for an unnecessary replication.

5. The authors have used the term "general population" in their title and text to define their target population: however, they have gone on to include studies of high-risk or vulnerable populations such as the mentally ill, those with epilepsy, pregnant women, and healthcare workers! There is little justification for introducing so much additional heterogeneity into what is already a very diverse and heterogeneous group of general population studies, especially since these groups already have higher prior rates of psychological distress. The study would be significantly improved in quality by a focus on general population studies alone.

6. In terms of improving the quality of the Discussion, see point #4 above.

Reviewer #2: The review explores the prevalence of mental health difficulties during the Covid-19 pandemic and the factors that are associated with such outcomes. MThe review has relevance and can make a contribution to the literature. My main issues are that the discussion should acknowledge weaknesses in the data and analysis. Principally that country of residence or nationality has not been included as a factor and that the mental health outcomes cannot be reliably attributed to Covid-19 from the data. I'd like to see the authors take up these issues and deal with them robustly in the discussion. More detailed comments are seen below.

1. There is no analysis of international region, surely there is variability here in the extent to which mental health difficulties arise given the wide variation in political and economic conditions in the many countries affected by Covid-19. Why did the authors not think that this was an important factor to consider in their analyses?

2. The measures used in the studies were mainly self-report questionnaires and this raises the possibility of demand characteristics and over-reporting of symptoms: what do the authors have to say about these potential confounds? The results suggest that 1 in 3 respondents have anxiety or depression, this is consistent with the Luo et al. (2020) meta-analysis as you say, but their review sampled studies which used the same types of self-report measures. This does not invalidate the reviews but it does present a very real weakness to claims about prevalence.

3. On page 22, last paragraph, you make the point about how age effects could be moderated by previous experience of infectious outbreaks. This is another reason why incorporating respondents’ country of residence is important, as some countries in Asia for example will have already experienced the SARS virus and might have better mental health outcomes as a result of this.

4. The discussion point raised on page 23 about how rural areas have poorer healthcare infrastructure, economy, sanitation, and education is contentious. In many countries the reverse might be true, whereby urban areas offer poorer environments/infrastructure.

5. The paper reports prevalence of mental health outcomes, but there is no way of linking such outcomes to the effects of Covid-19 itself. How do the authors address this issue and how does it affect the confidence of the conclusions they draw?

6. PLOS authors have the option to publish the peer review history of their article (what does this mean?). If published, this will include your full peer review and any attached files.

Reviewer #1: **Yes: **Ravi Philip Rajkumar

Reviewer #2: No

---

## [Author Response · Author response to Decision Letter 0]

1 Dec 2020

We would like to thank the editor and reviewers for their comments in improving the quality of our manuscript. The response letter has been uploaded as a word document, please kindly refer to the document. We hope that you are satisfied with our responses.

---

## [Decision Letter · Decision Letter 1]

15 Dec 2020

Factors associated with p sychological distress during the coronavirus disease 2019 (COVID-19) pandemic on the predominantly general population: a systematic review and meta-analysis

PONE-D-20-26693R1

Dear Dr. Jafar,

We’re pleased to inform you that your manuscript has been judged scientifically suitable for publication and will be formally accepted for publication once it meets all outstanding technical requirements.

Kind regards,

Michio Murakami

Academic Editor

PLOS ONE

Additional Editor Comments (optional):

Reviewers' comments:

Reviewer's Responses to Questions

**Comments to the Author**

1. If the authors have adequately addressed your comments raised in a previous round of review and you feel that this manuscript is now acceptable for publication, you may indicate that here to bypass the “Comments to the Author” section, enter your conflict of interest statement in the “Confidential to Editor” section, and submit your "Accept" recommendation.

Reviewer #1: All comments have been addressed

Reviewer #2: All comments have been addressed

2. Is the manuscript technically sound, and do the data support the conclusions?

Reviewer #1: Yes

Reviewer #2: Yes

3. Has the statistical analysis been performed appropriately and rigorously? 

Reviewer #1: Yes

Reviewer #2: Yes

4. Have the authors made all data underlying the findings in their manuscript fully available?

Reviewer #1: Yes

Reviewer #2: Yes

5. Is the manuscript presented in an intelligible fashion and written in standard English?

Reviewer #1: Yes

Reviewer #2: Yes

6. Review Comments to the Author

Reviewer #1: The authors have addressed the concerns raised in my earlier review.

I have no further major suggestions to make regarding further improvement or correction of this paper.

Reviewer #2: The authors have made credible responses to the issues raised in my first review and I am happy that the manuscript is sufficiently improved to be considered for publication by the editor.

7. PLOS authors have the option to publish the peer review history of their article (what does this mean?). If published, this will include your full peer review and any attached files.

Reviewer #1: **Yes: **Ravi Philip Rajkumar

Reviewer #2: No

---

## [Editor Report · Acceptance letter]

17 Dec 2020

PONE-D-20-26693R1 

Factors associated with psychological distress during the coronavirus disease 2019 (COVID-19) pandemic on the predominantly general population: a systematic review and meta-analysis 

Dear Dr. Jafar:

I'm pleased to inform you that your manuscript has been deemed suitable for publication in PLOS ONE. Congratulations! Your manuscript is now with our production department. 

Kind regards, 

on behalf of

Dr. Michio Murakami 

Academic Editor

PLOS ONE